# Catalytic Depolymerization of Date Palm Waste to Valuable C5–C12 Compounds

**Emmanuel Galiwango** [1,*] , **Ali H. Al-Marzuoqi** [1,*], **Abbas A. Khaleel** [2] **and Mahdi M. Abu-Omar** [3]

1 Department of Chemical and Petroleum Engineering, United Arab Emirates University, Al Ain 15551, United Arab Emirates

2 Department of Chemistry, United Arab Emirates University, Al-Ain 15551, United Arab Emirates; abbask@uaeu.ac.ae

3 Department of Chemistry and Biochemistry and Biochemistry, UC Santa Barbara, Santa Barbara, CA 93106, USA; mabuomar@ucsb.edu

* Correspondence: emmag@uaeu.ac.ae (E.G.); hassana@uaeu.ac.ae (A.H.A.-M.); Tel.: +971-3713-6407 (E.G.); +971-3713-4470 (A.H.A.-M.)

**Abstract:** Lignin depolymerization often requires multiple isolation steps to convert a lignocellulose matrix into high-value chemicals. In addition, lignin structural modification, low yields, and poor product characteristics remain challenges. Direct catalytic depolymerization of lignocellulose from date palm biomass was investigated. Production of high value chemicals heavily depends on optimization of different parameters and method of conversion. The goal of the study was to elucidate the role of different parameters on direct conversion of date palm waste in a bench reactor, targeting valuable C5–C12 compounds. The catalytic performance results demonstrated better liquid yields using a commercial alloy catalyst than with laboratory-prepared transition metal phosphide catalysts made using nickel, cobalt, and iron. According to the gas chromatography-mass spectrometry results, C7–C8 compounds were the largest product fraction. The yield improved from 3.6% without a catalyst to 68.0% with a catalyst. The total lignin product yield was lower without a catalyst (16.0%) than with a catalyst (76.0%). There were substantial differences between the carbon distributions from the commercial alloy catalyst, supported transition metal phosphide catalyst, and catalyst-free processes. This may be due to differences between reaction pathways. Lab-made catalysts cracked the biomass to produce more gases than the alloy catalyst. The final pressure rose from 2 bar at the start of the experiment to 146.15 bar and 46.50 bar after the respective reactions. The particle size, solvent type, time, temperature, gas, and catalytic loading conditions were 180 μm, methanol, 6 h, 300 °C, nitrogen, and 5 wt%, respectively. The results from this study provide a deep understanding of the role of different process parameters, the positive attributes of the direct conversion method, and viability of date palm waste as a potential lignocellulose for production of high-value chemicals.

**Keywords:** depolymerization; lignocellulose; direct conversion process; catalysis; high-value chemicals

## 1. Introduction

The increasing need for more renewable and sustainable fuels (liquid, solid, and gas) and high-value chemicals has motivated researchers to incorporate biomass-based feedstocks into the conventional petroleum-based infrastructure. In addition, the serious environmental pollution and finite nature of petroleum continue to motivate biomass-focused research [1,2]. Waste biomass can be processed to produce liquid fuels, including valuable oxygenates, for fuel additives and high-value chemicals [3–5]. Lignocellulosic biomass is a promising future raw material for fuels and high-value chemicals due to its carbohydrate (cellulose and hemicellulose) and lignin constituents [6,7]. The economics and processing technologies that are appropriate for the carbohydrate portion of this material are well understood. Numerous fuel additives and high-value products have been produced from this cellulose-rich component [7–10]. However, similar results have

not been achieved via fractionation of lignin into useful high-value products. Lignin is an aromatic polymer that consists of methoxylated phenyl propane units comprising three basic monolignols (sinapyl, coniferyl, and p-coumaryl alcohols). These exist in lignin molecules as syringyl, p-hyroxyphenyl, and guaiacyl [11–13]. Lignin accounts for approximately 15–40% of the lignocellulose biomass by dry weight and energy, and has the potential to produce sustainable, renewable high-value chemical phenolic platforms and fuels [14–17]. However, processing of lignin fractions from the lignocellulose complex in a biorefinery remains challenging [17]. Unlike cellulose, which has a single repeating β-1,4 glycosidic bond linkage, lignin has many complex β-O-4/4′, α-O-4/4′, β-β, β-5, and other linkages in its aromatic C-C and C-O-C bond centers [18,19]. The heterogeneity and varying nature of lignin make characterizing it a challenging task. In addition, lignin is resistant to many chemical treatments, and its conversion products are highly acidic and prone to repolymerization and condensation reactions. The above challenges limit the rapid development of lignin but offer many opportunities for further research designed to advance knowledge of lignin depolymerization and increase biorefinery competitiveness. Currently, lignin is used widely in the pulp and paper industry [20–22]. However, lignin research has attracted increasing attention and produced increasing advances towards production of second-generation bio-based fuels and high-value chemicals [23–25]. Among the many lignin conversion technologies such as pyrolysis, gasification, hydrothermal liquefaction, etc., selective depolymerization has attracted enormous interest as reported in various literature surveys [26–29]. The most common selective depolymerization strategy adopted by researchers for its robust and easily implemented approach is lignocellulose valorization to produce lignin. This is followed by catalytically (or non-catalytically) converting it to various chemical platforms and biofuels [30–34]. There are some drawbacks, such as low lignin yields and isolation efficiencies, fractionation techniques, and differences between biomass sources [35–37]. In addition, the commercial profitability of such a process based on a single lignocellulose component is difficult to predict in a biorefinery setting.

To counter some of the above challenges and to increase biorefinery competitiveness, this study sought to develop a single-step process approach to delignifying complex lignin and holocellulose (cellulose and hemicellulose) materials under various conditions to produce valuable, stable, high-value chemical precursors. The objectives of the study are (1) to investigate the roles of different parameters such as catalyst type, solvents, biomass particle sizes, reaction temperatures, reaction durations, and gas media, for their effects on production of C5–C12 compounds mainly from the lignin fraction of date palm waste biomass. (2) Investigate the potential of direct catalytic conversion route to high-value chemicals, and (3) investigate the potential of date palm biomass generated as a waste towards production of valuable high value chemicals similarly produced from second-generation biomasses. Target high-value chemicals in this range (C5–C12), such as cresols, are currently derived from petroleum and coal resources that are known to cause environmental pollution. In addition, cresols and their derivatives are widely applied in the food industry (vitamin E and other dietary supplements), agriculture, electronic encapsulation, chemical intermediates, etc. [38]. With surging demand for electronics like computers, smart phones, etc., the global market size for cresol is anticipated to grow from the 580 million US dollars recorded in 2015 [38]. Guaiacol, 2,6-dimethoxy-4,2-propenyl phenol, 2-methoxy-4-propylphenol, phenol, and other materials are useful bases or intermediate chemical platforms for applications in fuels, fragrances, agricultural materials, and many other industries [29,39]. This research is also motivated by the abundance of lignocellulose waste. Each year, approximately 500,000 tons of this waste is generated from about 44 million date palm trees in the United Arab Emirates, where this research was conducted. This waste is currently underutilized. Some of it is used in low-value addition applications such as compost, art, and crafts. In some instances, the waste is burnt and causes pollution, although this practice is not accepted by the government. Our previous study reported the non-isothermal kinetics and thermodynamic parameters for the pyrolysis of different date palm parts. Results showed high energy content and volatile

matter combined with low energy barriers, which makes date palm waste a potential candidate in a biorefinery [40]. Moreover, little has been reported on catalytic thermo conversion of date palm waste to high value chemicals. However, literature is available on biological transformation to high value chemicals [41], pyrolysis and characterization of pyrolytic properties [42], feedstock for biomass power plant to generate electricity and distilled water [43], and biochar production for several applications [44]. To make the biorefinery more competitive, more studies are needed to provide sufficient knowledge on catalytic thermo conversion of this waste to high value products.

Based on the information reported above, this paper reports production of high-value chemicals from a lignocellulose waste material via catalytic depolymerization process. The production process is robust and safe and eliminates multistep biomass pre-treatment stages. The results are fundamental to developing a competitive, less energy intensive biorefinery system that produces stable, valuable oxygenates with multiple applications from raw lignocellulose biomass.

## 2. Results and Discussion

### 2.1. Catalyst and Biomass Characterization

The biomass characteristics are shown in Table 1. Table 2 shows synthesized Brunauer-Emmett-Teller (BET) catalyst surface areas after calcination and reduction, as characterized via nitrogen adsorption at 77.35 K. As shown in Table 2, ɣ-$Al_2O_3$ has the largest surface area, but impregnation with metal phosphides reduces this surface area. This is attributed to metal oxide clusters produced during calcination and agglomeration of some phosphide species to block some of the pores in the ɣ-$Al_2O_3$ support. In addition, a further decrease in surface area is observed after thermal reduction [45]. This may be because the high annealing temperature used to reduce the crystalline metal oxides to metals decreases the pore diameters and reduces the pore volume. The commercial catalyst exhibits surface properties that are different from those of transition metal phosphide catalysts (TMPCs). For instance, the commercial catalyst has a much smaller surface area and larger pore size than the lab-prepared catalysts.

**Table 1.** Biomass elemental analysis (% on a dry, ash-free basis) and contents (wt%).

| C | H | O | Lignin | Holocellulose | Extractives | HHV [MJ/kg] | Ash | Moisture |
|---|---|---|---|---|---|---|---|---|
| 40.6 | 5.1 | 45.6 | 26.4 | 45.0 | $16.0 \pm 0.6$ | 15.0 | $8.7 \pm 1.2$ | $6.72 \pm 0.4$ |

HHV is the higher heating value.

**Table 2.** BET (Brunauer-Emmett-Teller) properties of the support and catalysts ([b] [46], [a] [47], [c] [48]).

| Entry | Catalyst Name | $S_{BET}$ [$m^2$ $g^{-1}$] | | Pore Volume [$cm^3$ $g^{-1}$] | Pore Size [nm] |
|---|---|---|---|---|---|
| | | Calcination | TPR | | |
| 1 | ɣ-$Al_2O_3$ | - | 258 | 0.39 [a] | 5.39 [a] |
| 2 | Commercial NiAl | - | 0.72 [b] | 0.000954 [b] | 53.3 [b] |
| 3 | $Ni_2P$/ɣ-$Al_2O_3$ | 200.8 | 172.6 | 0.26 [a] | 4.31 [a] |
| 4 | CoP/ɣ-$Al_2O_3$ | 153.7 | 106.1 | 0.43 [c] | 7.67 [c] |
| 5 | $Fe_2P$/ɣ-$Al_2O_3$ | 243.9 | 145.6 | 0.45 [a] | 5.74 [a] |

After the calcination process shown in Figure 1, in which nitrates and chlorides in the metal complexes are converted into metal oxides prior to temperature-programmed reduction (TPR) treatment, 2.0 g catalyst precursors were exposed to hydrogen TPR (50 mL/min) up to 950 °C at a ramp rate of 10 °C/min. Figure 2 shows the catalyst TPR results. The presence of more than one peak confirms the presence of more than one oxide in the catalyst bulk. Reduction of the Ni-based phosphide catalyst produces two distinctive peaks. The first peak appears at lower temperatures (500 °C) and signifies reduction of $NiPO_x$ and other nickel oxides deposited on ɣ-$Al_2O_3$ [49]. The higher-temperature reduction peak

is characteristic of other phosphate groups such as $PO_4^{3-}$, $(PO_3^-)_n$, and $P_2O_7$, whose reduction occurs at temperatures above 700 °C [50,51]. The two iron-based phosphide catalyst peaks are associated with the presence of iron oxide that is easily reduced at low temperatures. The peaks noted at higher temperatures may form because of reduction of iron phosphate and other components [52]. The cobalt phosphide catalyst exhibits one reduction peak. This indicates that few oxides are present in the bulk catalyst.

X-ray diffraction (XRD) diffractograms were used to confirm the presence of crystal lattices and metal oxides in the prepared catalyst bulk. Figure 3 shows XRD diffraction patterns for each as-deposited catalyst on the gamma alumina support. The $Ni_2P$ patterns (see Figure 3b) are observed at diffraction angles of 40.6°, 44.5°, and 66.2° and correspond to (111), (201), and (310) diffraction lines, respectively [53,54]. Diffraction from oxides of nickel (NiO) and phosphate-containing compounds ($AlPO_4$) is observed at 2θ values of approximately 18.6°, 22.0°, and 20.2°, 42.2°, 45.3°, respectively. The CoP diffraction peaks in Figure 3c exhibit 2θ values of 18.6°, 31.2°, and 57.2°, while the oxide $Co_3O_4$ peaks appear at 31.8° and 35.4°, as reported in the literature [55]. Oxides of iron (Figure 3d) are observed at 2θ = 22.3°, 33.1°, 42.3°, and 68.9°. These oxides are present in the form of $Fe_2O_3$ prior to calcination and are then successively reduced to $Fe_3O_4$ and FeO, which are finally reduced to metallic Fe. This phenomenon was reported in the earlier studies of calcination of the same catalyst at different temperatures during hydrodeoxygenation of 2-furyl methyl ketone as a model compound [47]. The phosphide iron peaks appear at 2θ = 15.0°, 40.2°, 42.0°, 57.6°, and 60.0° [56]. The high-surface-area Ɣ-$Al_2O_3$ exhibits strong diffraction peaks at 2θ values of 37.6° and 66.9° [45]. As shown in Figure 3a, the XRD patterns of the Ni-Al alloy reflect the presence of $Ni_3Al$ and $Ni_3Al_2$ domains. The largest commercial catalyst peak intensity appears at a diffraction angle of 44.7° and is attributed to the (111) reflection of the fcc alloy.

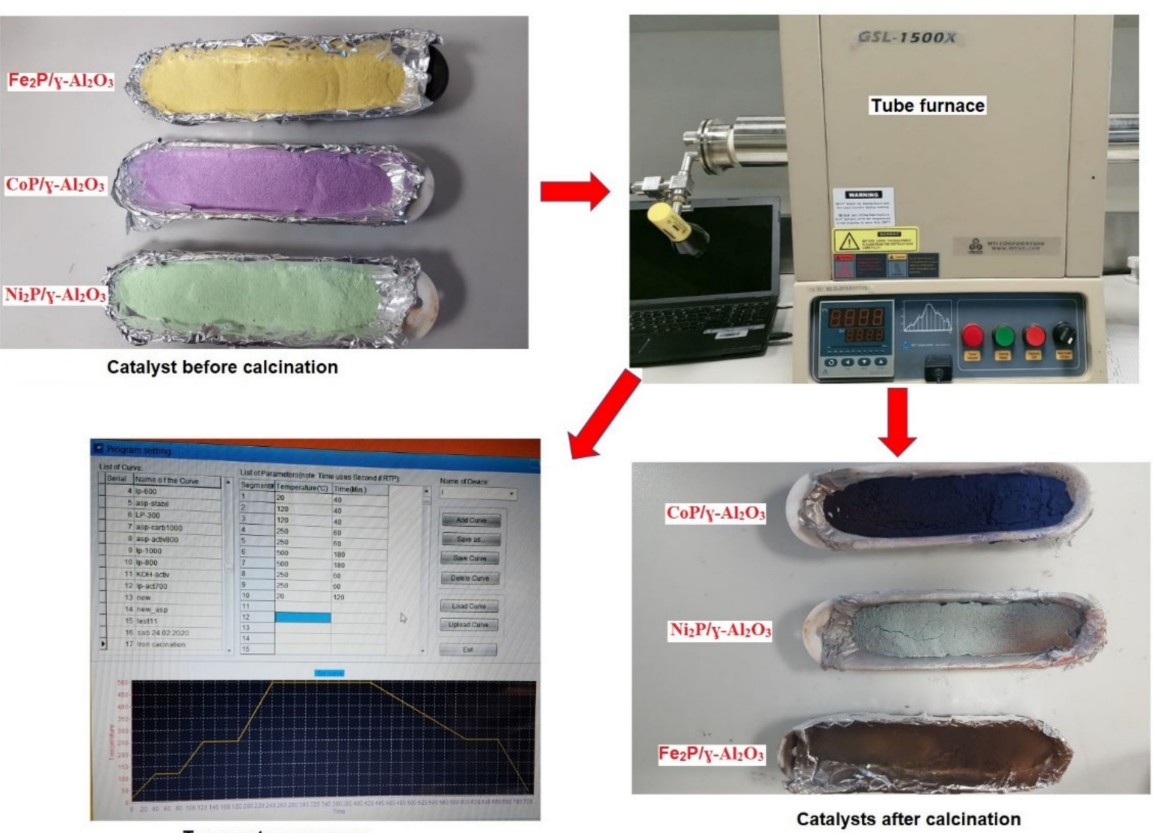

**Figure 1.** Calcination of $Ni_2P$/Ɣ-$Al_2O_3$, CoP/Ɣ-$Al_2O_3$, and $Fe_2P$/Ɣ-$Al_2O_3$ supported catalysts.

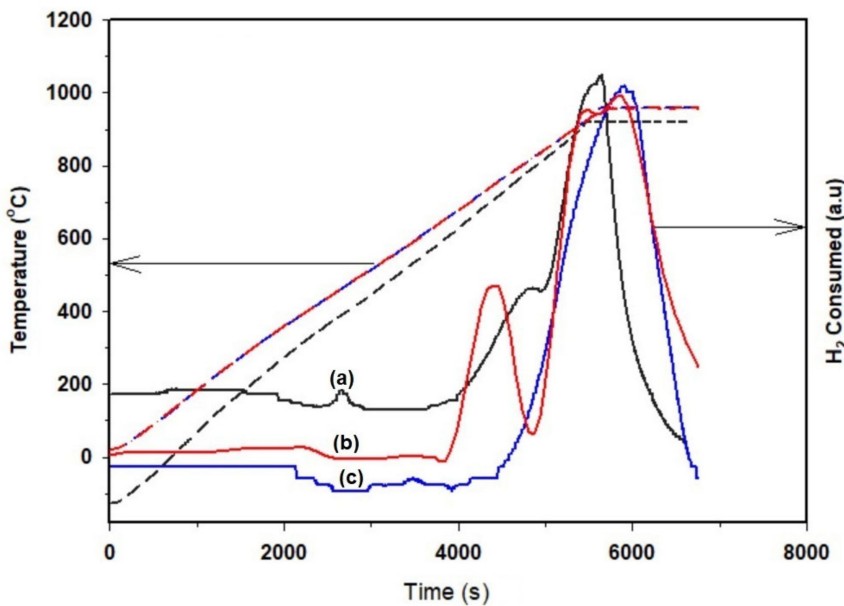

**Figure 2.** H$_2$-TPR profiles of 5 wt.% (**a**) Ni$_2$P/$\Upsilon$-Al$_2$O$_3$, (**b**) CoP/$\Upsilon$-Al$_2$O$_3$, and (**c**) Fe$_2$P/$\Upsilon$-Al$_2$O$_3$ supported catalysts.

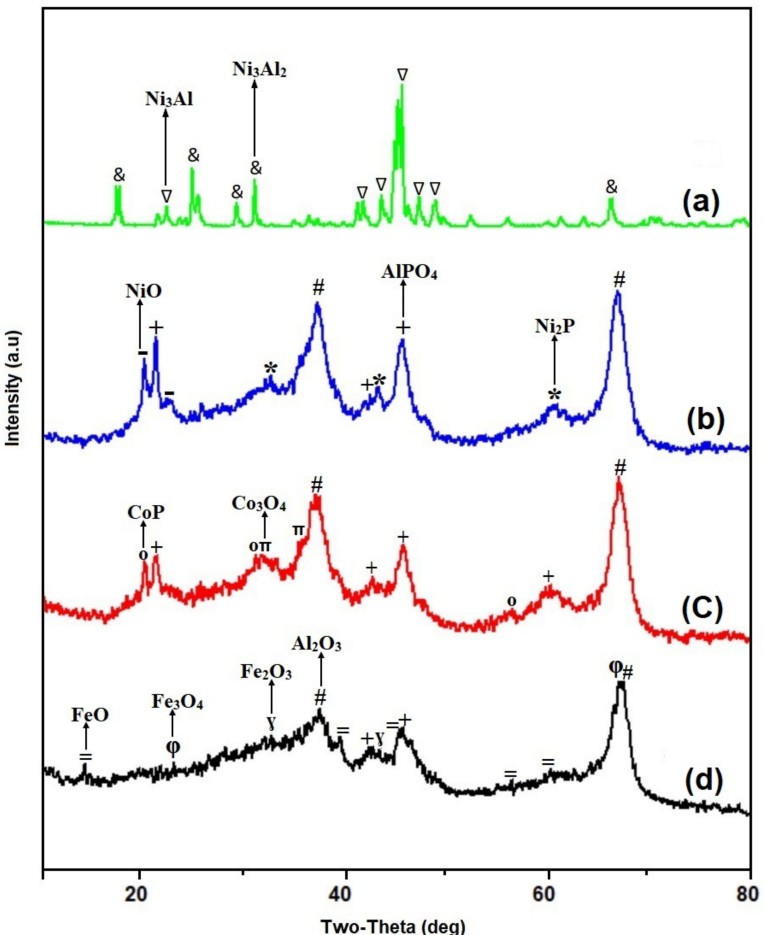

**Figure 3.** XRD (X-ray diffraction) patterns of 5 wt.% catalysts of (**a**) Commercial NiAl alloy, (**b**) Ni$_2$P/$\Upsilon$-Al$_2$O$_3$, (**c**) CoP/$\Upsilon$-Al$_2$O$_3$, and (**d**) Fe$_2$P/$\Upsilon$-Al$_2$O$_3$.

The Fourier transform infrared spectroscopy (FTIR) spectra of calcined and $H_2$-TPR samples are shown in Figure 4. Different functional groups are observed after the two treatment processes. After hydrogen TPR treatment, the catalysts exhibit decreased functional group intensities. This is attributed to high-temperature annealing and reduction of the oxides formed during calcination [57]. Typical Ɣ-$Al_2O_3$ spectra and their interactions as reported by most literature are assigned as follows. The bands with wave numbers, 3404, 3089, and between 1100–900 cm$^{-1}$ are attributed to O-H deformation vibrations, whereas the wavenumbers of 1639 and 1072 cm$^{-1}$ correspond to H-O-H stretching and asymmetric bending of Al-OH bonds, respectively [58–60]. Commercial NiAl catalysts exhibit similar functional groups at 1639 cm$^{-1}$. This is characteristic of H-O-H bond stretching from physically adsorbed water molecules [61]. However, the trend is different at higher wavenumbers, with spectra around 3050 cm$^{-1}$ assigned to carbonate ions that arise from interaction with the water molecule's hydrogen bonding during calcination preparation [62].

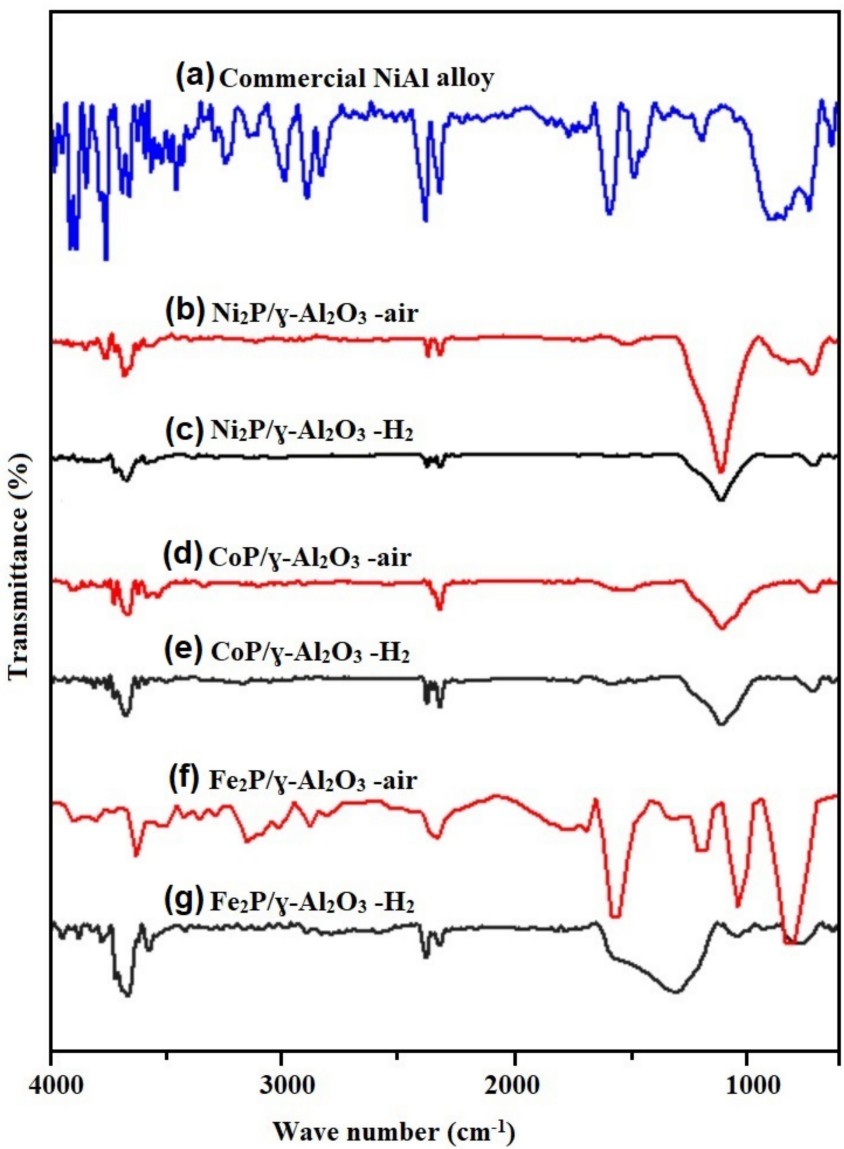

**Figure 4.** The Fourier transform infrared spectroscopy (FTIR) spectra of (**a**) commercial NiAl alloy, (**b**) $Ni_2P$/Ɣ-$Al_2O_3$-air, (**c**) $Ni_2P$/Ɣ-$Al_2O_3$-$H_2$, (**d**) CoP/Ɣ-$Al_2O_3$-air, (**e**) CoP/Ɣ-$Al_2O_3$-$H_2$ (**f**) $Fe_2P$/Ɣ-$Al_2O_3$-air, and (**g**) $Fe_2P$/Ɣ-$Al_2O_3$-$H_2$, catalysts after calcination (red colored) and reduction (black colored) treatments.

Figure 5 shows scanning electron microscopy (SEM) images of (a) $Ni_2P/Y-Al_2O_3$ and (b) $CoP/Y-Al_2O_3$ (c) $Fe_2P/Y-Al_2O_3$, and (d) commercial NiAl alloy catalysts. All catalysts exhibit rough surfaces and uneven morphologies. However, the nickel and iron-phosphide catalysts exhibit some even-shaped top surfaces, as marked in Figure 5 (a and c), respectively. Similar nickel and iron-based catalyst morphologies have been reported in the literature [63,64]. The commercial Ni-Al exhibits intact morphologies similar to those reported in other works [65].

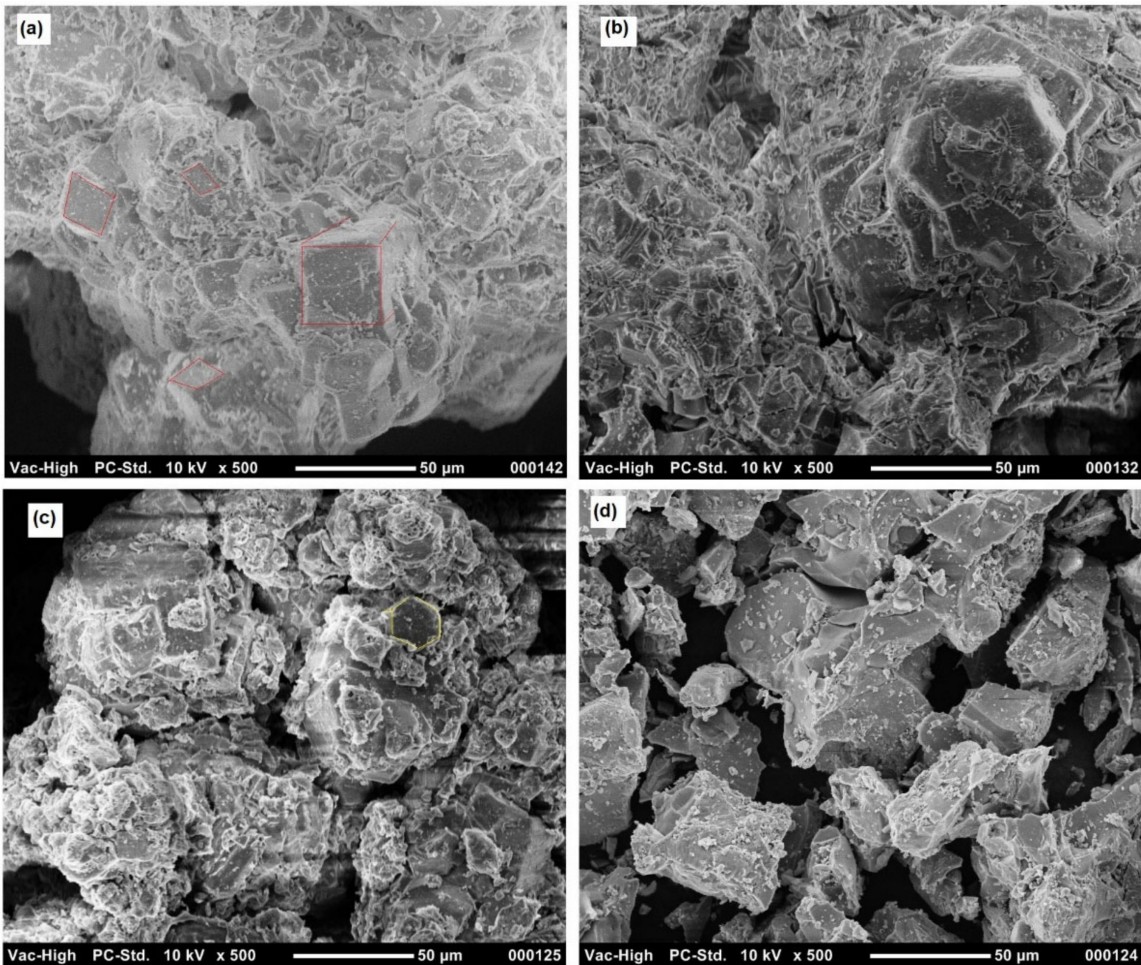

**Figure 5.** SEM (scanning electron microscopy) images of (**a**) $Ni_2P/Y-Al_2O_3$ and (**b**) $CoP/Y-Al_2O_3$ (**c**) $Fe_2P/Y-Al_2O_3$, and (**d**) commercial NiAl alloy catalysts at X500.

### 2.2. The Effects of Reaction Parameters on High-Value Product Distributions

The catalytic effect of biomass fractionation was investigated using a commercial NiAl alloy catalyst and selected laboratory-prepared transition metal phosphide catalysts. As shown in Table 3, the commercial alloy and transition metal phosphide catalyst product distributions after solvent recovery are different. This suggests that there might be differences in reaction pathways. In addition, differences in catalyst properties, such as the surface area and crystal lattice and reducible oxide contents, may lead to different reaction pathways when the substrate is accessed. TMPC-based reactions and reactions without catalysts exhibit higher gas yields than reactions performed using the NiAl alloy catalyst. The highest liquid yield is (51.5 wt%). This can be attributed to production of non-condensable vapors, formation of low molecular weight compounds during catalytic cracking of biomass, and possible hydrodeoxygenation (HDO) reactions [66]. Moreover, TMPCs are widely used in HDO and gasification reactions that lead to evolution of more

gases than liquids [67,68]. High value chemical (HVC)-containing liquid yields were analyzed via gas chromatography-mass spectrometry (GC-MS) for product selectivity. NiAl alloy catalysts with higher liquid yields were used as base catalysts for screening of various reaction parameters.

**Table 3.** Date palm waste biomass product distribution in a bench reactor [b].

| Catalysts | | NiAl Alloy | Ni$_2$P/ɣ-Al$_2$O$_3$ | CoP/ɣ-Al$_2$O$_3$ | Fe$_2$P/ɣ-Al$_2$O$_3$ | Without Catalyst |
|---|---|---|---|---|---|---|
| Product yield (wt. %) | Solid | 14.1 ± 0.6 | 22.4 ± 0.7 | 23.5 ± 0.4 | 18.8 ± 0.7 | 32.0 ± 0.2 |
| | Liquid | 51.5 ± 1.0 | 24.6 ± 1.5 | 26.9 ± 0.9 | 22.6 ± 1.0 | 16.6 ± 0.4 |
| | Gas | 34.4 ± 0.4 | 53.0 ± 0.2 | 49.6 ± 0.2 | 58.6 ± 0.2 | 51.4 ± 0.2 |

[b] Reaction conditions: Time 6 h, temperature 300 °C, catalyst loading 5 wt%, 2 bars initial N$_2$ pressure, methanol 75 mL, biomass 3.7 g, 20 rpm.

The effect of the reaction temperature on product yields at various reaction times was investigated and is shown in Figure 6. The results in Figure 6a–c show that at temperatures below 240 °C, the product yield of C5–C12 compounds is low even after a prolonged reaction time of 6 h. Under the same conditions, the product distribution includes few C5–C12 compounds at reaction times of 4 h and 6 h. Most of the yield is distributed among other products. These are primarily products from conversion of the holocellulose portion of the feedstock. The results from the GC-MS analysis revealed some of the holocellulose-derived compounds such as furfural, hydroxymethyl-furfural, arabinose, methyl $\alpha$-D-glucopyranoside. Higher temperatures between 240 °C and 300 °C favor production of more C5–C12 compounds, with the highest yields observed after 6 h of reaction time. Similar optimal phenolic derivative production reaction temperatures and times have been reported in the literature [69,70]. The quantified C5–C12 compounds include "furfural (C5), furfural alcohol (C5), phenol (C6), m-cresol (C7), guaiacol (C7), methyl benzoate (C8), isoeugenol (C10), 2-methoxy-4-propylphenol (C10), 5-isopropyl-2-methylphenol (C10), 2,6-dimethoxy-4,2-propenyl phenol (C11) and 3,4-diethyl-2,4-hexadienedioic acid, dimethyl ester (C12)".

The effect of catalytic loading on delignification is shown in Figure 7. The catalytic performance of the NiAl alloy catalyst is better at 5 wt% than at higher catalytic loading levels of 25 wt% and 50 wt%. Reports of lower catalyst dosages producing higher delignification yields are present in the literature. For instance, Huang and team used 0.5–3 g of a Ru/SiC catalyst with 10 g of apple-wood biomass. Low catalyst dosages produced the highest extent of delignification and a product that contained 47 wt% monomer [71]. In addition, Kim and team reported lower performance of direct formic acid fuel cell with higher catalyst loading which was explained to be due the thick catalyst layer that might have inhibited the proton transport [72]. Furthermore, during the deoxygenation studies of m-cresol on Pt/γ-Al$_2$O$_3$, low catalytic loading was reported to give high yields of light hydrocarbons [73]. However, generally an increase in catalyst loading increases the process performance, it is worthy to note that the increment is not always linear, but rather there is an optimal catalyst loading for each specific reaction. Beyond that optimal value regardless of the how much the catalytic loading is increased, the productivity either increases or remains constant [72,74]. Regardless of the catalytic loading, the NiAl catalyst has better C7–C8 compound activity. In addition, a catalytic loading of 25 wt% produces better C5–C6, C9–C10, and >C10 product yield distributions. The results show that the product distribution can vary depending on the availability of catalytic active sites on the substrate and the ratio of catalyst to biomass.

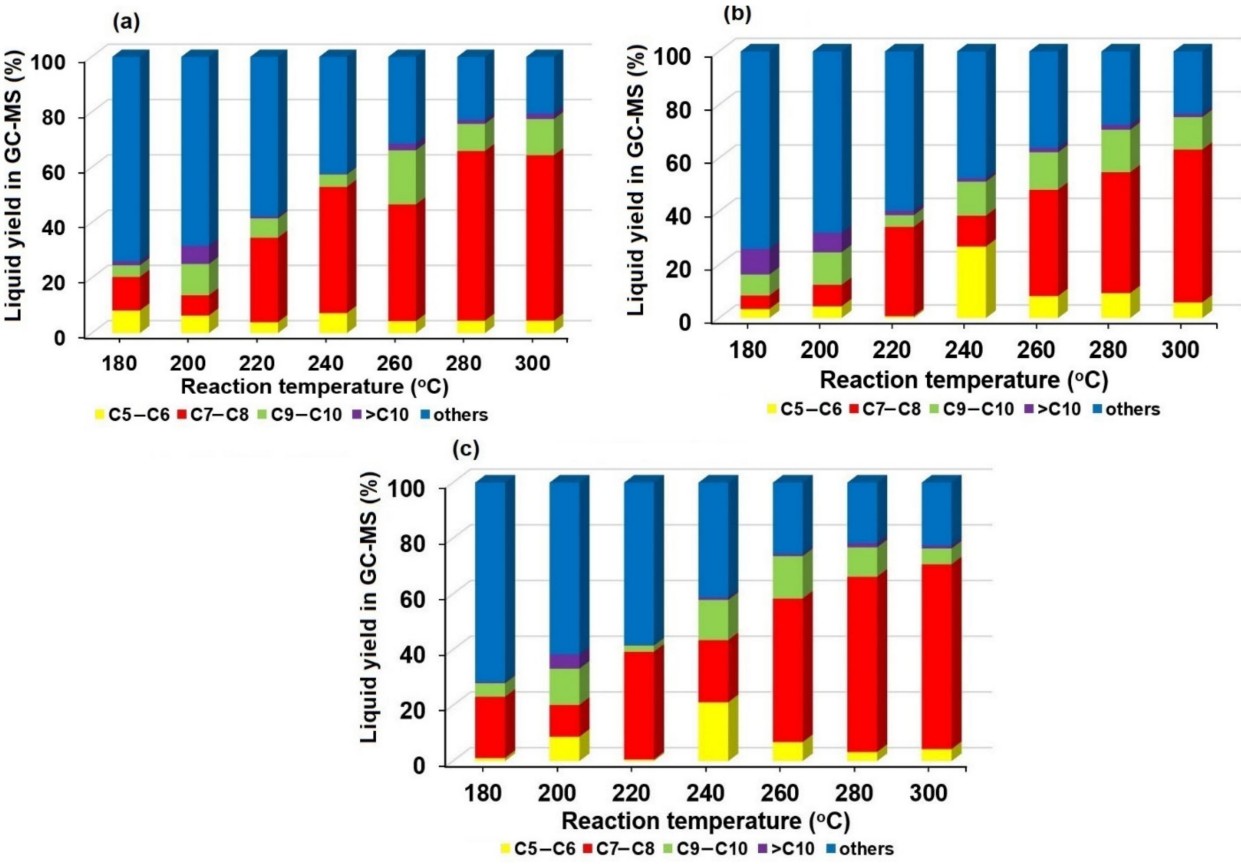

**Figure 6.** The effect of the temperature on the product liquid yield after (**a**) 2 h, (**b**) 4 h, and (**c**) 6 h of reaction time using a 5 wt. % NiAl catalyst, 2 g biomass, 2 bar initial pressure, and reaction temperature between 180–300 °C.

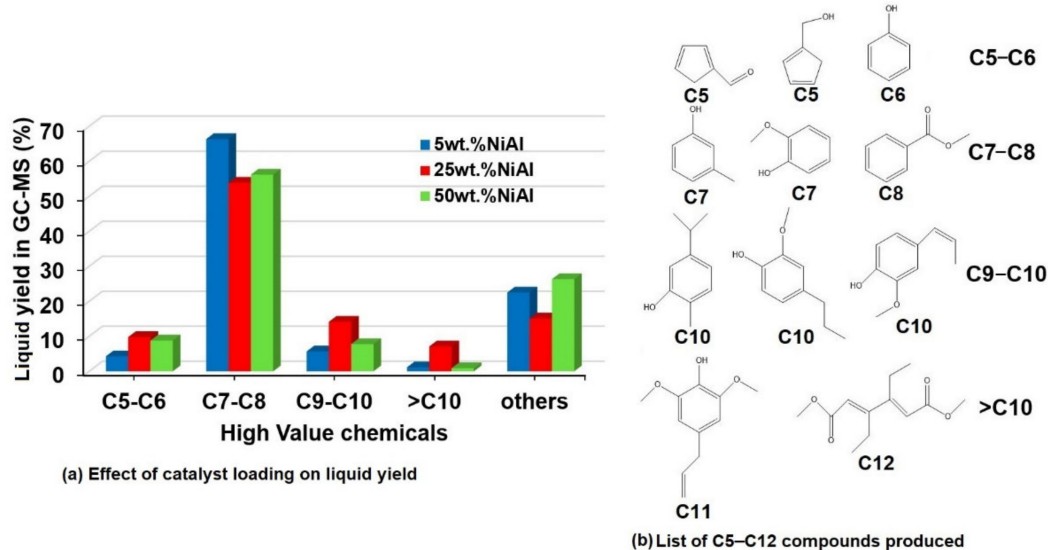

**Figure 7.** Effect of catalytic loading with respect to biomass on liquid yields at reaction conditions of 300 °C, 6 h, 2 g biomass, 2 bar initial pressure, methanol as solvent, and biomass particle size of 180 μm.

Various solvent media such as water, ethanol, propanol, and their mixtures have been used for lignin depolymerization because they provide oxidative or reductive media for catalytic biomass fractionation [75]. Low mono-aromatic oxygenate yields such as 6.2–6.9 wt% (with water as solvent) improve to 7.1–7.9 wt% upon switching to ethanol [75]. Recent findings have suggested that methanol can improve the yield from conversion

of lignin to HVC. For example, Luo and co-workers depolymerized Miscanthus lignin to 69 wt% phenolics using methanol in the presence of hydrogen and a reaction time of 12 h [76]. Matson and group attempted one-step biomass catalytic conversion using supercritical methanol under harsh conditions of 160–220 bar and 320 °C [77]. Inspired by the ability of methanol to produce better yields, while keeping in mind the cost and safety risks posed by the methods shown above, this study investigated the effects of safe, inexpensive solvents on conversion of waste biomass to HVCs. As shown in Figure 8, all the solvents studied have the potential to fractionate date palm waste biomass to various HVCs. Methanol exhibits the highest C7–C8 compound product yield of 67.95%. In addition, water and dioxane produce relatively good yields of C5–C6.

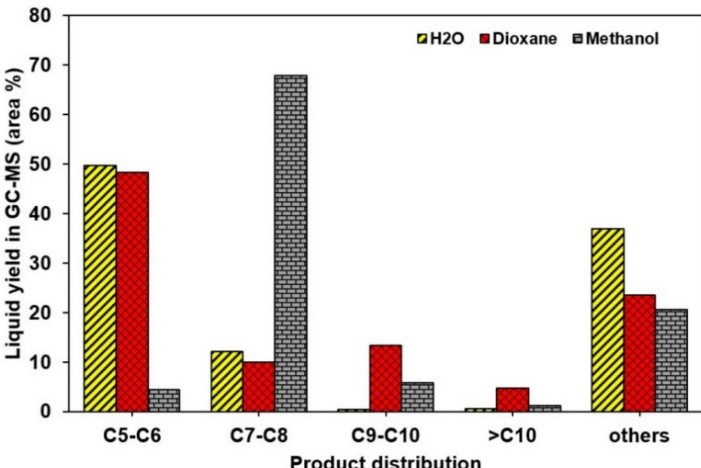

**Figure 8.** Effect of solvent on the high value chemical (HVC) liquid fraction yield after 6 h at 300 °C under 2 bars initial $N_2$ pressure with 5 wt% commercial NiAl, 2 g biomass, methanol solvent, and biomass particle size of 180 μm.

The effect of biomass particle size was investigated as shown in Figure 9. The results suggest that the investigated particle size range has little effect on the product distribution. This implies that all particle sizes offer the effective heat transfer required for the reaction. C7–C8 HVCs are highly selective at all particle sizes.

The optimal reaction parameters produced by catalytic screening using commercial NiAl are 300 °C, 6 h, 180 μm, methanol, and 5 wt% for the temperature, time, particle size, solvent, and catalytic loading with respect to the amount of biomass, respectively. The highest selectivity was to m-cresol (a constituent of C7–C8 HVCs) with a yield of 68.0%, as shown in Figure 8. The production of highly selected HVCs from date palm waste was investigated across different reaction times and temperatures using the same NiAl alloy catalyst as shown in Figure 10. At lower reaction time (2 h), m-cresol yields showed no definite trend with increasing temperature. However, when the reaction time was increased to 4 h, m-cresol yields increased with the temperature, except at 240 °C, which seems to suggest stability in m-cresol production with prolonged reaction time. Furthermore, after 6 h reaction time, a similar trend of increase in m-cresol yield with the temperature was observed at temperature >240 °C. M-cresol is often produced from biomass as a model compound during catalytic upgrading [78,79]. However, a few studies have reported production of m-cresol from lignocellulose biomass. For instance, Zhang and team reported m-cresol selectivity of 6.31 mol% via microwave pyrolysis catalytic conversion using ZSM-5 [80]. In addition, the concentration of m-cresol in treated lignin with calcium oxide increased to about 200% in a study of calcium catalyzed pyrolysis of lignocellulose biomass components [81]. M-cresol is commonly used in soap emulsions as a disinfectant, and is a major precursor during synthesis of chemical intermediates for other compounds and materials including pesticides, plastics, dyes, and pharmaceuticals [82].

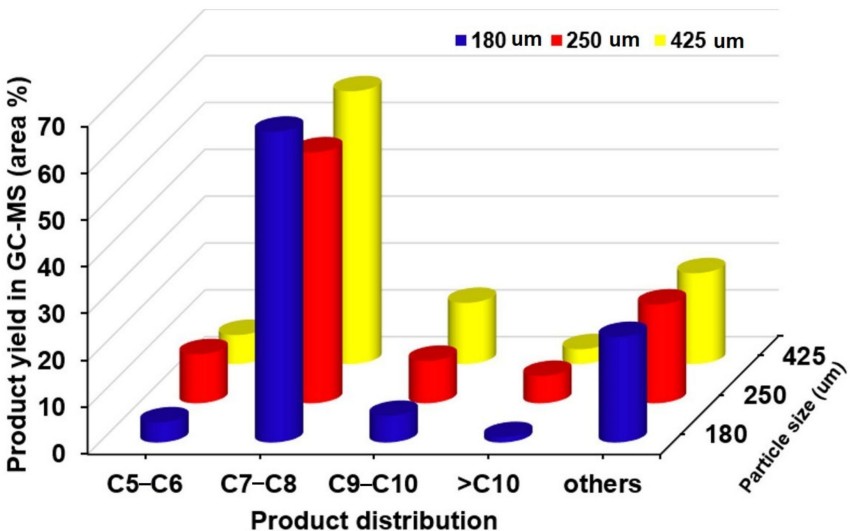

**Figure 9.** The effect of the biomass particle size on the HVC liquid fraction yield after 6 h at 300 °C under 2 bars initial $N_2$ pressure with 5 wt.% commercial NiAl with respect to biomass amount, 2 g biomass, and methanol solvent.

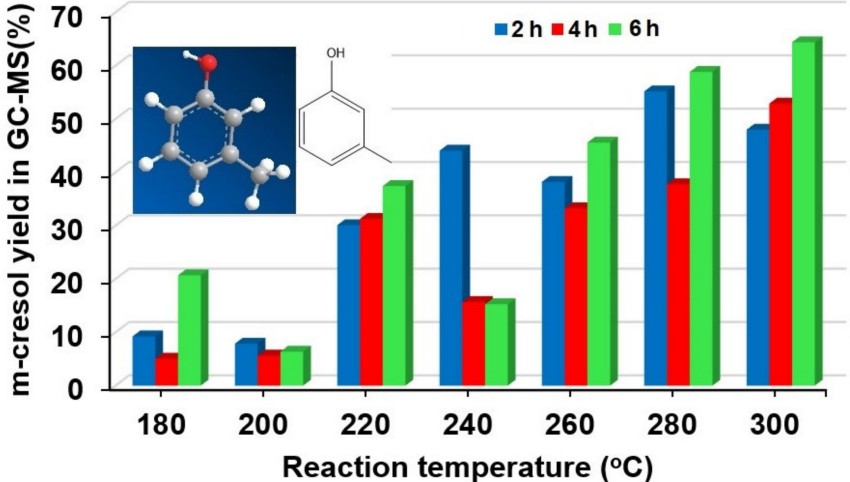

**Figure 10.** The effect of the reaction temperature on the m-cresol yield at various reaction times using 5 wt.% commercial NiAl catalyst with respect to biomass quantity at reaction conditions of 2 bars initial $N_2$ pressure, 2 g biomass, methanol solvent, and 180 μm biomass particle size.

Reductive fractionation of biomass to HVCs has often been conducted using hydrogen as the reducing gas in the presence of a catalyst. A few reports have tried and suggested lignin depolymerization in the absence of hydrogen while utilizing the peculiar properties of hemicelluloses and solvents to provide the protonation needed for catalytic reduction [34]. In this work, the role of nitrogen in the presence of methanol was investigated for both commercial alloy catalysts and TMPCs. The proposed reaction pathways of complex lignocellulose structure (lignin, cellulose, and hemicellulose) (Figure 11) involved the lignin decomposition mainly into phenolic derivatives and holocellulose into C5 carbon compounds. The results in Figures 12 and 13 show liquid product yields and HVC distributions. Biomass fractionation yields more liquid in the presence of $N_2$ than $H_2$ (Figure 12). This is attributed to peculiar hemicellulose properties and hydrogen transfer hydrogenolysis from the organic reductive methanol solvent [83]. Similar literature results showed more reactive hydrogen sources than addition of ultra-pure hydrogen for lignin depolymerization [84]. However, the gas medium seems to have no definite effect on the HVC product distribution. In addition, TMPCs produce

low yields and a wide range of C5–C12 compounds (highest yield in phenol), unlike the NiAl alloy catalyst, which is highly selective for C7–C8 compounds (with the highest yield in m-cresol). This suggests a difference between the reaction mechanisms of the two catalyst groups because of differences in their surfaces and morphological structures, as shown in the characterization section. Moreover, TMPCs produce more gas-phase products, just as in the literature reports where they were used for gasification of biomass [85]. The gas-phase products were not analyzed by GC-MS in this study, but rather the total quantity of gas produced per run was determined from mass balance. The trend of product distribution for TMPC was highly selective for high molecular compounds with best in carbon number > C10 and poorly selective for low molecular compounds of C5–C6. It is worth noting that product distribution for TMPCs was hugely dependent on the type of gas used during reaction. The trend of product distribution in the presence of $N_2$ gas was in decreasing order of >C10 > C9–C10 > C5–C6 > C7–C8. However, the trend changed in the presence of $H_2$ to C5–C6 > C9–C10 > >C10 > C7–C8 with low molecular compounds best selected. This difference in product distribution owing to change in the type of gas media used can be postulated to the difference in extent of each gas's participation in catalytic reduction during biomass fractionation. A summary of analysis results that reflect the selected reaction conditions is displayed in Table 4. The results in Table 4 show that commercial NiAl alloy catalyst lower final $N_2$ pressure than TMPCs. This signifies that commercial NiAl alloy catalyst limits the conversion of delignified fractions to gaseous phase by converting them more to liquid products as opposed to TMPCs. Regardless of the solvent used, both catalyst categories produced similar compounds. However, commercial NiAl alloy catalyst had the best selectivity. Furthermore, both catalysts pose ability to improve the products quality through improved pH values exhibited when catalysts were used. The conversion rate was highest for commercial NiAl alloy catalyst (7.34 $h^{-1}$) at 2 h reaction time. The conversion rate trend for commercial NiAl alloy catalyst increased linearly with temperature and decreased linearly with reaction time. In addition, there was increased conversion at higher temperatures, which seem to suggest a possible increase in turnover at specified conditions. The strong dependence of yield on temperature was also reported in literature and supports the idea that heat may be underexplored in overwhelming product inhibition in catalysis [86].

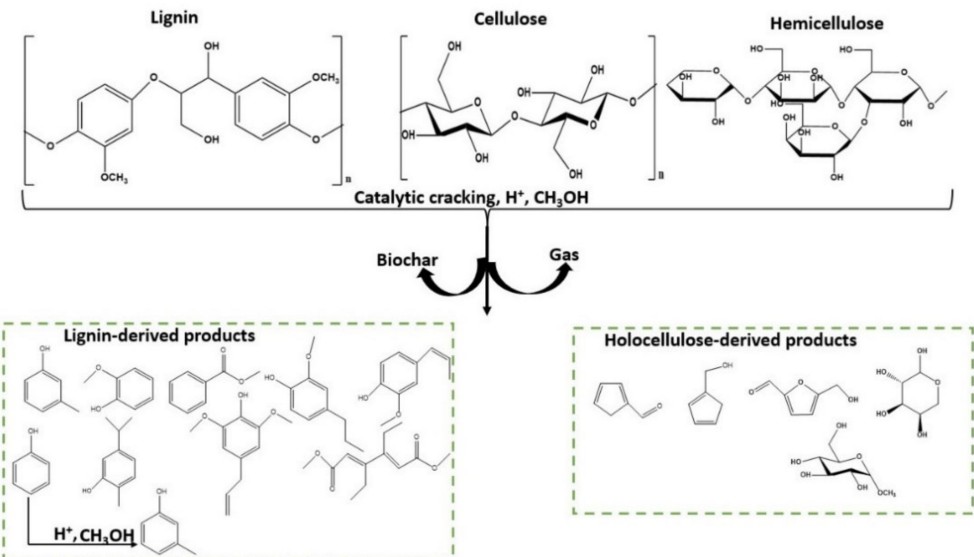

**Figure 11.** The possible reaction pathways for conversion of date palm waste into high value chemicals.

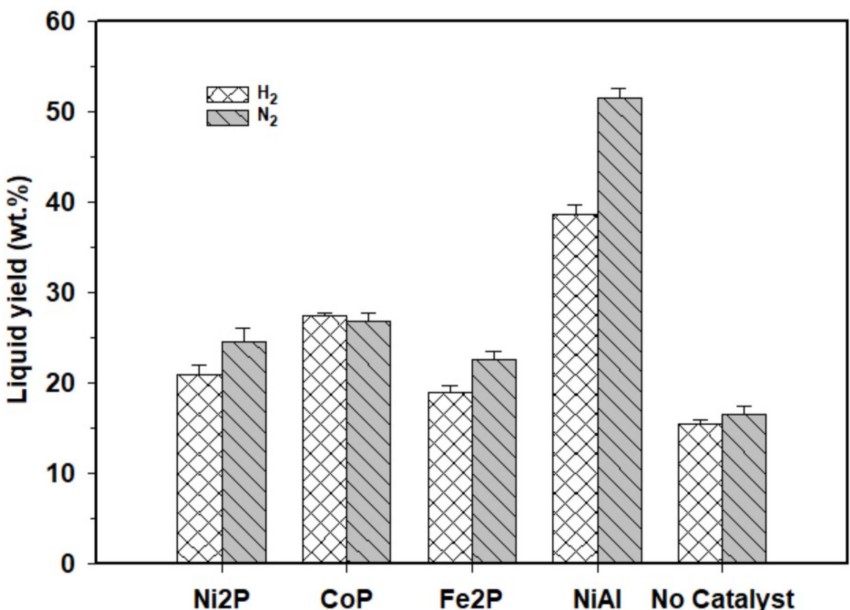

**Figure 12.** The effects of various gas media ($N_2$ and $H_2$) on liquid yields produced after 6 h at 300 °C, 2 bars initial pressure, 5 wt% catalyst loading with respect to biomass amount, 2 g biomass, methanol solvent, and 180 μm biomass particle size.

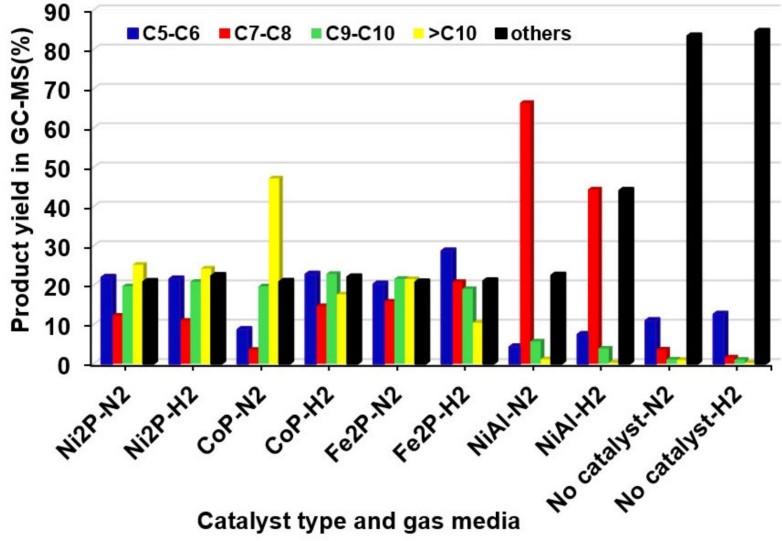

**Figure 13.** The effects of various catalysts and gas media ($N_2$ and $H_2$) on HVC yields at reaction conditions of 6 h, 300 °C, 2 bars initial pressure, 5 wt.% catalyst loading with respect to biomass amount, 2 g biomass, methanol solvent, and 180 μm biomass particle size.

It is worthy to note that determination of plausible reaction mechanism for the natural lignin remains a challenge due to lignin complex linkages and highly reactive intermediates that occur during oxidation and reduction process [87]. Thus, much attention regarding reaction mechanism is being studied mainly on model compounds.

**Table 4.** Summary of date palm fractionation analysis results from a reactor charged with 2 bars initial $N_2$ pressure, 5 wt.% catalyst loading with respect to biomass amount, 2 g biomass, and 180 µm biomass particle size.

| Catalyst | Solvent | T [°C] | $P_{N2}$ after Reaction [bar] | Time [h] | Conversion * [%] | pH | Compound Name and Formula | Liquid selectivity GC-MS [%] | Liquid Yield [%] | Rate ** [h$^{-1}$] |
|---|---|---|---|---|---|---|---|---|---|---|
| NiAl alloy | methanol | 180 | 16.3 | 2 | 26.0 ± 0.5 | 5.42 | m-cresol ($C_7H_8O$) | 35.5 | 9.2 ± 0.2 | 2.48 |
| NiAl alloy | methanol | 200 | 25.3 | 2 | 31.6 ± 0.2 | 5.05 | Isoeugenol($C_{10}H_{12}O_2$) | 24.9 | 7.9 ± 0.1 | 2.82 |
| NiAl alloy | methanol | 220 | 28.4 | 2 | 42.3 ± 1.1 | 5.39 | m-cresol ($C_7H_8O$) | 71.1 | 30.1 ± 0.8 | 3.53 |
| NiAl alloy | methanol | 240 | 30.4 | 2 | 57.8 ± 3.6 | 5.69 | m-cresol ($C_7H_8O$) | 76.3 | 44.1 ± 2.7 | 5.61 |
| NiAl alloy | methanol | 260 | 33.1 | 2 | 68.6 ± 1.1 | 5.42 | m-cresol ($C_7H_8O$) | 55.7 | 38.1 ± 0.6 | 6.56 |
| NiAl alloy | methanol | 280 | 38.3 | 2 | 77.2 ± 0.3 | 5.40 | m-cresol ($C_7H_8O$) | 71.5 | 55.2 ± 0.2 | 6.76 |
| NiAl alloy | methanol | 300 | 42.5 | 2 | 79.5 ± 0.4 | 5.47 | m-cresol ($C_7H_8O$) | 60.4 | 48.0 ± 0.2 | 7.34 |
| NiAl alloy | methanol | 180 | 16.8 | 4 | 25.9 ± 0.7 | 5.48 | 2,6-dimethoxy-4,2-propenyl phenol, DMPP ($C_{11}H_{14}O_3$) | 19.5 | 5.1 ± 0.1 | 1.26 |
| NiAl alloy | methanol | 200 | 25.7 | 4 | 32.0 ± 0.7 | 5.10 | 2-methoxy-4-propylphenol DHE ($C_{10}H_{14}O_2$) | 17.6 | 5.6 ± 0.1 | 1.52 |
| NiAl alloy | methanol | 220 | 35.7 | 4 | 40.3 ± 1.5 | 5.37 | m-cresol ($C_7H_8O$) | 77.6 | 31.3 ± 1.2 | 1.85 |
| NiAl alloy | methanol | 240 | 46.3 | 4 | 52.4 ± 0.4 | 5.60 | Phenol ($C_6H_6O$) | 29.9 | 15.7 ± 0.1 | 2.50 |
| NiAl alloy | methanol | 260 | 46.5 | 4 | 63.9 ± 0.6 | 5.28 | Phenol ($C_6H_6O$) | 26.8 | 17.1 ± 0.2 | 3.06 |
| NiAl alloy | methanol | 280 | 44.1 | 4 | 72.5 ± 0.7 | 5.33 | m-cresol ($C_7H_8O$) | 52.1 | 37.8 ± 0.4 | 3.43 |
| NiAl alloy | methanol | 300 | 40.6 | 4 | 76.7 ± 0.5 | 5.31 | m-cresol ($C_7H_8O$) | 69.0 | 52.9 ± 0.4 | 3.50 |
| NiAl alloy | methanol | 180 | 16.8 | 6 | 28.4 ± 1.6 | 5.45 | m-cresol ($C_7H_8O$) | 72.8 | 20.7 ± 1.1 | 0.83 |
| NiAl alloy | methanol | 200 | 24.65 | 6 | 38.3 ± 2.2 | 5.09 | Isoeugenol($C_{10}H_{12}O_2$) | 16.6 | 6.4 ± 0.4 | 1.21 |
| NiAl alloy | methanol | 220 | 34.3 | 6 | 41.8 ± 2.8 | 5.41 | m-cresol ($C_7H_8O$) | 89.6 | 37.4 ± 2.5 | 1.33 |
| NiAl alloy | methanol | 240 | 37.7 | 6 | 58.8 ± 1.9 | 5.65 | m-cresol ($C_7H_8O$) | 26.0 | 15.3 ± 0.5 | 1.68 |
| NiAl alloy | methanol | 260 | 49.5 | 6 | 74.4 ± 2.2 | 5.38 | m-cresol ($C_7H_8O$) | 61.2 | 45.6 ± 1.4 | 2.33 |
| NiAl alloy | methanol | 280 | 44.6 | 6 | 78.4 ± 1.2 | 5.39 | m-cresol ($C_7H_8O$) | 75.1 | 58.8 ± 0.9 | 2.55 |
| NiAl alloy | methanol | 300 | 43.6 | 6 | 77.5 ± 0.5 | 5.68 | m-cresol ($C_7H_8O$) | 83.1 | 64.4 ± 0.4 | 2.58 |
| No catalyst | methanol | 300 | 7.5 | 6 | 16.6 ± 1.2 | 3.94 | Phenol ($C_6H_6O$) | 50.5 | 8.4 ± 0.6 | - |
| NiAl alloy | 1,4-dioxane | 300 | 55.6 | 6 | 76.4 ± 0.9 | 3.63 | Phenol ($C_6H_6O$) | 41.6 | 31.8 ± 0.4 | 2.21 |
| NiAl alloy | Water | 300 | 77.73 | 6 | 63.0 ± 0.6 | 3.51 | Phenol ($C_6H_6O$) | 54.0 | 34.00 ± 0.3 | 1.89 |
| Ni$_2$P/Ɣ-Al$_2$O$_3$ | methanol | 300 | 140.6 | 6 | 79.1 ± 1.4 | 4.77 | 3,4-diethyl-2,4-hexadienedioic acid, dimethyl ester (HDD-ester) | 20.5 | 16.2 ± 0.3 | 2.56 |
| CoP/Ɣ-Al$_2$O$_3$ | methanol | 300 | 136.1 | 6 | 79.0 ± 2.5 | 4.86 | 3,4-diethyl-2,4-hexadienedioic acid, dimethyl ester (HDD-ester) | 22.6 | 17.8 ± 0.6 | 2.57 |
| Fe$_2$P/Ɣ-Al$_2$O$_3$ | methanol | 300 | 146.15 | 6 | 79.2 ± 2.1 | 4.82 | Phenol ($C_6H_6O$) | 35.9 | 28.4 ± 0.8 | 2.50 |

* Conversion is weight of total liquid products after solvent removal over weight of biomass fed to the reactor × 100. ** Rate is the conversion rate calculated as reactant weight consumed/weight of catalyst/h.

## 3. Materials and Methods

### 3.1. Sample and Catalyst Preparation

The date palm waste biomass used in the study was provided by the University farm in Al foah Al Ain, UAE in August 2019. The samples were pruned from mature (>10 years) date palm trees and then cleaned of farm dirt prior to size reduction using a high-speed electric grinder. The samples were sieved with the aid of an automatic shaker and various mesh sizes to produce samples with particle sizes of 180 μm, 250 μm, and 300 μm. The samples were dried in an oven at 105 °C overnight to equilibrate their moisture contents prior to further valorization. The biomass underwent gravimetrically determined extractive removal via Soxhlet extraction (48 h) with ethanol and benzene (1:2, *v/v*) [88]. The biomass molecular weight was 72 kDa and was characterized for proximate and ultimate analysis using various ASTM standards as described in our previous works [89,90].

A commercial nickel aluminum alloy (150 μm) catalyst was used to screen various reaction parameters. The optimal conditions were tested using laboratory-prepared transition metal phosphide catalysts. The commercial NiAl alloy catalyst was purchased from Sigma-Aldrich (product number: 221651) and used without further treatment. All the gamma alumina ($\gamma$-Al$_2$O$_3$) (97%, Aldrich, product number: 199966) supported Ni$_2$P, CoP, and Fe$_2$P metal phosphide catalysts were prepared via the incipient wetness impregnation technique. Metal loading of 5 wt% was used and a 1:1 phosphorus to metal (P:M) molar ratio was selected [91]. The desired amounts of nickel (II) chloride hexahydrate NiCl$_2$.6H$_2$O (98%, Aldrich), cobalt (II) nitrate hexahydrate Co(NO$_3$)$_2$·6H$_2$O (98%, Aldrich), and iron (II) nitrate nanohydrate Fe(NO$_3$)$_2$·9H$_2$O (98%, Aldrich), and the amount of ammonium phosphate dibasic ((NH$_4$)$_2$HPO$_4$) (98%, Aldrich) required to meet the above P:M ratios were calculated and the materials were dissolved successively in de-ionized water while stirring. Thereafter, citric acid (CA) (99.0%, Aldrich) was added to the salt solution in a 2:1 CA:M molar ratio to prevent aggregation of metal species and increase the surface area [92]. The homogeneous catalyst precursor solutions were impregnated into the $\gamma$-Al$_2$O$_3$ support (pore volume 0.39 cm$^3$ g$^{-1}$) and dried in an oven at 120 °C for 24 h. Example for the preparation of Ni$_2$P/$\gamma$-Al$_2$O$_3$ is as follows: 2.014 g of nickel (II) chloride hexahydrate (NiCl$_2$.6H$_2$O) were dissolved in 10 mL of distilled at room temperature followed by adding 1.119 g of ammonium phosphate dibasic ((NH$_4$)$_2$HPO$_4$). 3.254 g of citric acid (CA) were then added to the preceding thick solution under continuous stirring until the solution became green. Using a dropper, the suspension was deposited on a pre-weighed gamma alumina support (9.237 g) in another beaker. The wet impregnated catalyst was dried in an oven at 100 °C for 2 h prior to further characterization. Other laboratory prepared TMPCs were prepared using the same procedure following calculation of their corresponding precursor amounts.

The dried catalysts were sieved through a 90-micron mesh and 5 g calcined at 500 °C using a temperature ramp rate of 2 °C/min under constant air flow for 3 h in a tube furnace. After calcination, the catalysts were characterized via TPR, X-ray diffraction (XRD), scanning electron microscopy (SEM), Fourier transform infrared spectroscopy (FTIR), and the Brunauer–Emmett–Teller (BET) method. TPR analysis was conducted using ChemBET PULSAR TPR/TPD analyzer (Quantachrome Instruments, Florida, USA). The catalyst samples (0.05 g) with particle size of 174 microns were reduced under H$_2$ (10%) at atmospheric pressure at a flow rate of 50 mL/min. During TPR analysis, the reactor was heated to 960 °C using a heating rate of 2 °C/min. The H$_2$ consumption was measured as a signal using Quantachrome TPRWin software at a detector current of 150 mA. The crystallography of the reduced catalysts was analyzed using an X-ray diffractometer (PANalytical -XPERT-3, Philips, The Netherlands). The X-ray diffraction measurements were performed within the 2θ range of 10° to 80° using Cu K$\alpha$ X-ray source radiation energy with a voltage of 40 kV and current of 30 mA. Catalyst structural morphologies were analyzed via SEM (JEOL Neoscope JCM-5000, Tokyo, Japan). The samples were coated in gold using an Au/C vacuum sputtering device while clamped to the sample holder. The images were captured using a spot size of 40 at 10 kV. An IRTrace-100 FTIR

spectrophotometer (Shimadzu, Kyoto, Japan) was used for the ATR-FTIR analysis. The calcined and reduced catalysts were analyzed to determine the contribution of the support towards the redox properties of the metal oxide-supported catalysts. Spectral results were recorded within a range of 500 cm$^{-1}$ to 4000 cm$^{-1}$ using a 4 cm$^{-1}$ spectral resolution and 34 scans. The surface areas and pore sizes of the calcined and reduced catalysts were determined using the BET method (Novatouch LX2, Quantachrome Instruments, Boynton Beach, FL, USA).

### 3.2. Experimental Apparatus and Reaction Conditions

Conversion of lignocellulose date palm waste to high-value product precursors was performed in a high-pressure, high-temperature reactor (parr-4560, USA). As shown in Figure 14. the reactor assembly consisted of a temperature-(500 °C) and pressure (345 bar)-rated, 250 mL cylindrical vessel, a fixed head with various valve fittings, an impeller, a thermocouple, and other parts. The experiment was performed using various temperatures (180–300 °C), biomass particle sizes (180–425 μm), extents of catalyst loading (5–50 wt%), solvent types, reaction durations (2–6 h), and gas media. The reactions were performed in three replicates and the reactor was cooled to room temperature after each run. The liquid fractionates were separated from the solid phase with the aid of a 0.45 μm filter. The catalysts had magnetic properties and were separated from the unreacted solids using a magnetic bar. The solvents were recovered with the aid of a rotary evaporator. The product yields were calculated gravimetrically as a ratio of the products to the biomass fed to the system [48]. The gas yield was calculated as the difference between 100% and the sum of solids and liquids.

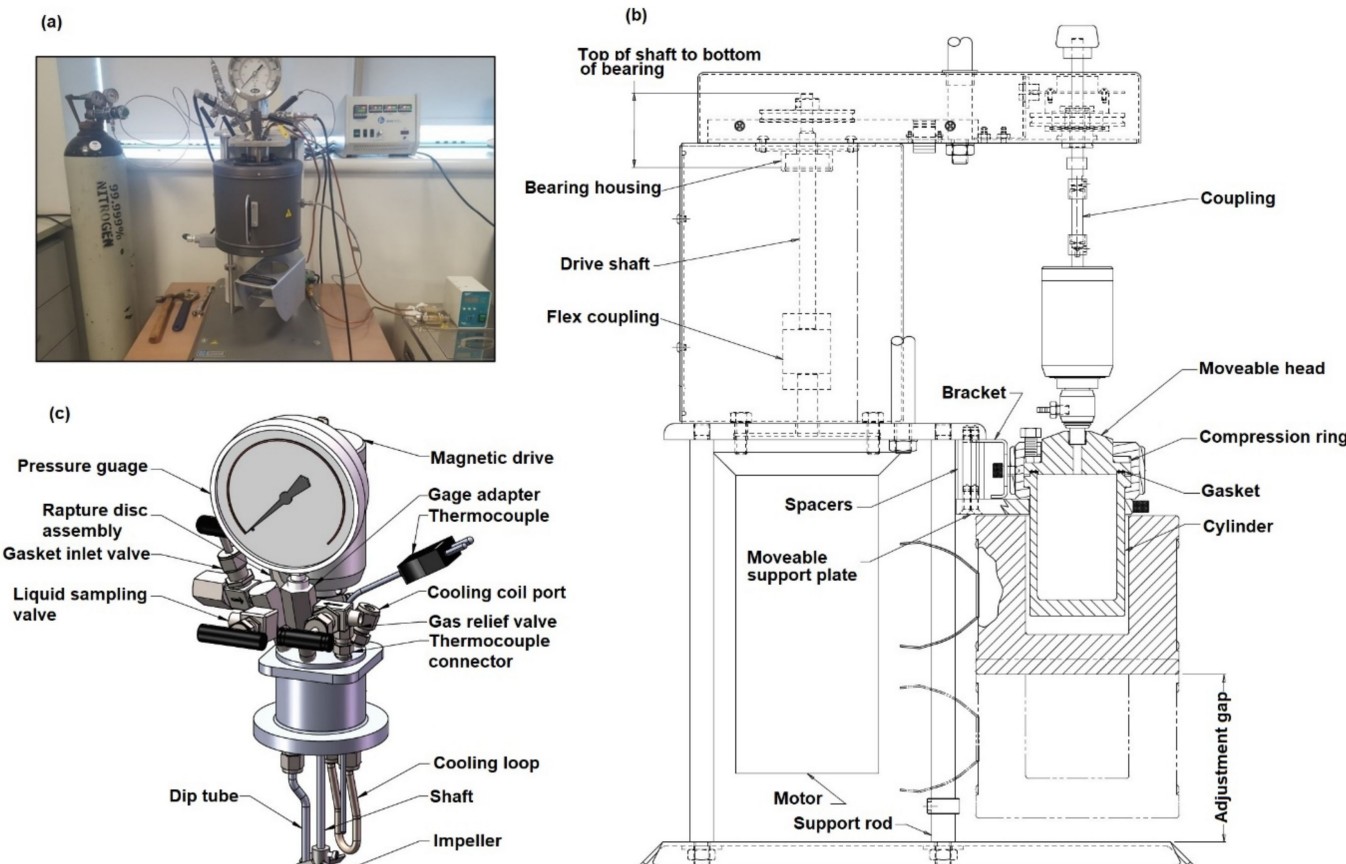

**Figure 14.** The high-pressure, high-temperature reactor (**a**) pictorial diagram, (**b**) assembly drawing, and (**c**) fixed head.

Qualification and quantification of the HVC-containing liquid product was performed in a GC-MS instrument (Thermo Scientific, GC trace 1300, and ISQ MS detector, WA, USA) using helium as the carrier gas. Quantification was performed using standards reported in the literature for the expected phenolic derivatives [70]. The TraceGOLD™ TG-WAXMS column (0.25 um Thickness; 0.32 mm ID; 30 m Length) was heated to 150 °C for 1 min and thereafter elevated to 250 °C at a heating rate of 20 °C min$^{-1}$. The latter condition was maintained for an extra 8 min before 1.0 μL of material was injected with the aid of an autosampler using a split ratio of 60:1. The component selectivity ($S_i$), conversion ($X_i$), and yield ($Y_i$) were calculated from Equations (1)–(3). To compare the number of reactant molecules converted per minute per catalytic weight ($0.1 \pm 0.05$ g) for given reaction conditions of the two catalyst groups used, conversion rate was calculated according to Equation (4).

$$S_i\ (\%) = \frac{\text{Integrated area of product i}}{\text{Total integrated area of all detected products}} \times 100 \tag{1}$$

$$X_i(\%) = \frac{\text{weight of total liquid products after solvent removal}}{\text{weight of the biomass fed to the reactor for a particular run}} \times 100 \tag{2}$$

$$Y_i = S_i \times X_i \tag{3}$$

$$\text{Conversion rate} = \frac{\text{weight of reactant consumed}}{\text{weight of catalyst x(time of reacton)}} \tag{4}$$

## 4. Conclusions

One step thermochemical lignin depolymerization of a lignocellulose matrix to high-value chemical products was investigated in a high-pressure, high-temperature reactor. The liquid product yield increased from $16.6 \pm 0.4$ wt% without a catalyst to 51.53 wt% with commercial NiAl alloy catalyst and highest transition metal phosphide catalyst of CoP/γ-Al$_2$O$_3$ with 26.91 wt%. Although the catalytic performance was best with the alloy NiAl catalyst, transition metal phosphide catalysts showed potential for direct lignin depolymerization from lignocellulose, which is different from the hydrodeoxygenation processes commonly used to produce compounds. The catalytic and solvent effect improved the liquid product quality and increased the pH from 3.94 without a catalyst to a maximum of 5.69 with a catalyst. With the alloy catalyst, the carbon distribution was primarily C7–C8, while the transition metal phosphide catalysts produced product distributions in order of C5–C6, C9–C10, and >C10 compounds. M-cresol had the best yield and selectivity of 67.95% and 89.61%, respectively. This study revealed the feasibility of single-step lignocellulose fractionation in a reactor, and the potential for use of alloy and transition metal phosphide catalysts to produce useful liquid oxygenates for energy and high-value chemical applications.

**Author Contributions:** Conceptualization, E.G. and A.H.A.-M.; methodology, E.G.; formal analysis and investigation, E.G., A.H.A.-M., A.A.K., and M.M.A.-O.; resources, A.H.A.-M. and A.A.K.; writing—original draft preparation, E.G.; writing—review and editing, E.G., A.H.A.-M., A.A.K., and M.M.A.-O.; supervision, A.H.A.-M., A.A.K., and M.M.A.-O.; project administration, A.H.A.-M.; funding acquisition, A.H.A.-M. and M.M.A.-O. All authors have read and agreed to the published version of the manuscript.

**Funding:** This research received no external funding.

**Data Availability Statement:** Not applicable.

**Acknowledgments:** This research was funded by UAEU Emirates Center for Energy and Environment (31R107)

**Conflicts of Interest:** The authors declare no conflict of interest.

## Abbreviations

| | |
|---|---|
| BET | Brunauer-Emmett-Teller |
| TPR | Temperature-programmed reduction |
| XRD | X-ray diffraction |
| SEM | Scanning electron microscopy |
| FTIR | Fourier transform infrared spectroscopy |
| TMPCs | Transition metal phosphide catalysts |
| HHV | Higher heating value |
| HDO | Hydrodeoxygenation |
| GC-MS | Gas chromatography-mass spectrometry |
| CA | Citric acid |
| ɣ-Al$_2$O$_3$ | Gamma alumina |
| HVCs | High value chemicals |

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
