# Peer review of "Catalytic Depolymerization of Date Palm Waste to Valuable C5–C12 Compounds"

_catalysts, doi:10.3390/catal11030371_

Round 1
Reviewer 1 Report
Manuscript review: „Catalytic depolymerisation of date palm waste to valuable C5-C12 compounds” (Manuscript ID: catalysts-1114137)
The manuscript deals with the production of high-value chemicals from lignocellulose waste by catalytic depolymerization. The aim of the research presented in the manuscript was to clarify the role of different parameters in the direct conversion of palm waste in a bench top reactor targeting valuable C5-C12 compounds.
The authors have made significant improvements. This should undoubtedly be appreciated. I don't notice substantive mistakes.
I would very much appreciate the addition of a table of all abbreviations in the manuscript. The page numbering in the manuscript is poorly done. Literature needs improved formatting.
Author Response
We appreciate the reviewer’s efforts to improve the quality of the manuscript.
We are delighted to report our responses in the attached file.

Reviewer 2 Report
The manuscript can be accepted in its present form.
Author Response
We appreciate the reviewer’s efforts to improve the quality of the manuscript.

Reviewer 3 Report
I reviewed the original manuscript (catalysts-1005654) and had several comments. Unfortunately, the authors did not replay to them – in “Itemized Replies to Reviewers’ Comments” my comments are missing. I read the new version of manuscript and I found that the authors´ action was not adequate.
- Reaction condition are still missing in legends to Figs 11, 12 and Table 4. In legend to Fig.9 particle size of 180 μm is given erroneously.
- The time and temperature dependence of m-cresol yield (Fig. 10) is strange. It really needs the better explanation.
- HHV abbreviation (Table 1) should be explained.
- “Most of the yield is distributed among other products. These are primarily products from conversion of the holocellulose portion of the feedstock.” Please, give examples of products or references.
- Table 4 does not seem trustworthy: a)To give values of conversions, yields and selectivities on two decimal points is meaningless. b) To compare catalyst effectiveness the authors newly added values of TOF. However, it is not standard to use TON or TOF in the case of depolymerization reaction. What value of molecular weight the authors took for reactant? The authors should have rather used productivities as yield of a product (weight or moles) per catalyst weight per unit time. c) If we compere conversion development in time (e.g. lines 7,14,21), conversion does not change practically, but TOF increases systematically. How is it possible?
- I agree with the recommendation of academic editor to present possible reaction pathways to form the observed products, because the authors claim in text as well in abstract “product distributions after solvent recovery are different. This suggests that there might be differences in reaction pathways.”
In summary, I do not recommend publication of manuscript without a thorough revision.
Author Response

(The authors gave the same response as above.)

Round 2
Reviewer 3 Report
The reaction of the authors on my last comments was adequate except:
- Fig.9 legend – please delete 180 μm particle size, because particle size is a variable,
- Discussion to Fig. 10 – I do not think that “the increase in m-cresol yield with temperature” can be called “exponential”,
- TOF values in Table 4 – I agree that TOF depends on catalyst concentration and that it can increase or decrease independently of the conversion achieved, however, in my example (experiments on lines 7,14,21) reaction conditions are the same, only reaction time increases. According to your eq.
TOF = number of moles of reactant consumed/moles of catalyst x time of reaction TOF values should decrease with increasing reaction time, because the amount of catalyst is the same, the amount of biomass is the same and its conversion is practically the same (76-79%). It seems that some important information concerning Table 4 and/or TOF calculation is missing.
Author Response
We appreciate the reviewer’s efforts to improve the quality of the manuscript.
All the minor reviews have been addressed as seen in the attached file
